# An Efficient and Cost-Effective Approach to Generate Functional Human Inducible Pluripotent Stem Cell-Derived Astrocytes

**DOI:** 10.3390/cells12192357

**Published:** 2023-09-26

**Authors:** Hemil Gonzalez, Srinivas D. Narasipura, Tanner Shull, Amogh Shetty, Tara L. Teppen, Ankur Naqib, Lena Al-Harthi

**Affiliations:** 1Department of Internal Medicine, Division of Infectious Diseases, Rush University Medical Center, Chicago, IL 60612, USA; 2Department of Microbial Pathogens and Immunity, Rush Medical College, Chicago, IL 60612, USA; srinivasa_narasipura@rush.edu (S.D.N.); tanner_l_shull@rush.edu (T.S.); 3Division of Epidemiology and Biostatistics, School of Public Health, University of Illinois, Chicago, IL 60608, USA; 4Illinois Mathematics and Science Academy, Aurora, IL 60506, USA; 5Molecular Neurobiology Division, Rush Alzheimer’s Disease Center, Rush University, Chicago, IL 60612, USA; 6Genome Core Facility, Rush University, Chicago, IL 60612, USA; ankur_naqib@rush.edu

**Keywords:** astrocytes, iAs, matrigel, iPSC, NPC, glutamate uptake, seahorse

## Abstract

Human inducible pluripotent stem cell (hiPSC)-derived astrocytes (iAs) are critical to study astrocytes in health and disease. They provide several advantages over human fetal astrocytes in research, which include consistency, availability, disease modeling, customization, and ethical considerations. The generation of iAs is hampered by the requirement of Matrigel matrix coating for survival and proliferation. We provide a protocol demonstrating that human iAs cultured in the absence of Matrigel are viable and proliferative. Further, through a side-by-side comparison of cultures with and without Matrigel, we show significant similarities in astrocyte-specific profiling, including morphology (shape and structure), phenotype (cell-specific markers), genotype (transcriptional expression), metabolic (respiration), and functional aspects (glutamate uptake and cytokine response). In addition, we report that, unlike other CNS cell types, such as neuronal progenitor cells and neurons, iAs can withstand the absence of Matrigel coating. Our study demonstrates that Matrigel is dispensable for the culture of human iPSC-derived astrocytes, facilitating an easy, streamlined, and cost-effective method of generating these cells.

## 1. Introduction

Astrocytes are a significant brain cell population. They are critical for maintaining brain homeostasis, including physical integrity and [1,2,3,4,5,6]. They comprise a pleomorphic population highly adaptive to local signals [7,8,9]. Astrocyte plasticity enables them to perform a variety of key functions, including the regulation of water and ion homeostasis, neurotransmitter recycling, tissue repair and chemotactic signaling, detoxification and removal of waste substances, and immune surveillance [10,11,12,13]. At the neurovascular interface, pericapillary astrocytes support the structure, development, and repair of the blood brain barrier (BBB) through their end-feet projections [2,3]. From an energetics perspective, the astroglial tricarboxylic acid cycle (TCA cycle) accounts for up to 14% of the brain’s oxygen consumption; this is due largely to its role in maintaining the glutamate-glutamine cycle between neurons and astrocytes [14,15]. This tight regulation of neurotransmitter concentrations at the synaptic cleft protects neurons from excitotoxicity [11]. In addition to their close interactions with neighboring non-astroglial cells, astrocytes interface with cellular networks by coupling their gap-junctions [16,17,18]. By communicating through these intercellular channels, astrocytes generate ultrastructural cytoplasmic continuity, resulting in a functional syncytium [19].

Research into the biology of astrocytes in health and disease requires a reliable and consistent supply of primary astrocytes. Typically, astrocytes are sourced from human adult brains, glioma cell lines, aborted human fetal tissues, or rodent brains [20,21,22,23,24]. Each source has its caveats. Aborted fetal tissues and human brain specimens are difficult to access and can be scarce. Further, human fetal astrocytes (HFAs) exhibit limited proliferative capacity, show batch-to-batch variation, and their transcriptome is different in comparison to human adult astrocytes [24,25,26]. Adult human astrocytes are typically retrieved from a non-diseased portion of a brain during surgery or postmortem; albeit, it is still a diseased brain [27]. Astrocytoma cell lines are immortalized and may not recapitulate the biology of healthy astrocytes [22,28]. Mice primary astrocytes, on the other hand, although easier to access, are laborious to isolate and show several transcriptional and phenotypic differences compared to adult primary human astrocytes [29]. Recently, human induced pluripotent stem cells (hiPSCs) were successfully differentiated into astrocytes (iAs) via neural progenitor cells (NPC) or by induction of global transcription factors such as SOX9 and NFIB directly in hiPSCs [30,31]. Other methods include neurosphere-mediated differentiation [32]. iAs are replicative, transcriptomically consistent, easily accessible, cost-effective, and can be generated on a larger scale [30,31]. As such, iAs are increasingly emerging as an important tool in studying astrocyte biology and function.

Organic matrices are biopolymeric hydrogels that serve as an extracellular matrix (ECM), which provides structural and nutritional support for adherent cells to attach and develop under ideal conditions, thus acting as the basal laminae found in native tissues in vivo [33]. Matrigel is derived from the Engelbreth-Holm-Swarm sarcoma and is the most used product for this purpose, but other products such as PLO/laminin are also used [34,35,36]. In the brain, the ECM provides a scaffold to support both neuronal and non-neuronal cells. Growth factors bind to ECMs and impact cell survival and proliferation [37]. Like iPSCs, most non-astrocytic brain cells, such as neurons, microglia, and oligodendrocytes, require ECM coating for their routine culture in vitro [38,39,40].

Because astrocytes are adherent cells, the current consensus in the field is that organic matrix coating with products such as Matrigel is an essential prerequisite for culturing them [30,31,41,42,43,44]. We assessed whether iAs can be cultured in vitro without Matrigel or PLO/laminin coating, which will be cost-effective and reduce the time required to generate hiPSC-astrocytes.

## 2. Results

The morphology, viability, proliferation, and phenotypes of iAs cultured without Matrigel highly resemble those of iAs cultured on Matrigel.

Astrocytes were derived from two human iPSC cell lines and one NPC line, as described [31]. Briefly, NPCs were plated on Matrigel-coated plates at low cell density (15,000 cells/cm^2^) at day 0. On day 1, NPC media were replaced with cAM. Cells were fed with cAM every other day and split every 10 days after that until day 60. Matrigel coating was used throughout this period, and after day 60, cells were cultured with and without Matrigel for further studies. iPSC lines exhibited industry standards for quality control by morphology (colonies with tight borders, Appendix A) and high expression of classical markers, OCT-4, and TRA-1-60 (Appendix A). All the NPC lines exhibited expected morphologies (80–90% confluency with tightly packed small spherical shaped cells, Appendix A) with high expression of classical markers, Nestin and PAX-6 (Appendix A). We assessed whether Matrigel coating is required for the generation of phenotypically and functionally active astrocytes. We cultured iAs generated from two iPSC lines and an NPC line side by side on tissue culture plates with and without Matrigel coating and evaluated morphology by light microscopy under phase contrast. We observed strong similarity in morphological features, including stellate shape (arrowheads), long cellular processes (arrows), and low overlap between processes in all iAs irrespective of matrix coating (Figure 1A and Appendix A).

Next, we assessed the viability of iAs in wells with and without Matrigel by detaching with accutase and counting attached cells after 72 h of culture. There was no significant difference between viable cells from coated and non-coated plates in RUCDR-iAs (Figure 1B) as well as in Axol-iAs (Appendix A) and ATCC-iAs (Appendix A). We then assessed the proliferation rate of RUCDR-iAs across multiple time points. We found no significant difference in proliferation rates between coated and non-coated cells (Figure 1C). Similar observations were made in Axol-iAs and ATCC-iAs (Appendix A).

Further, we evaluated the expression of Ki67, a key regulator of cell proliferation, and observed no statistically significant differences in Ki67 protein expression in RUCDR-iAs or Axol-iAs between Matrigel-coated and uncoated conditions (Figure 1D,E and Appendix A). Similarly, we observed no statistically significant difference in the transcription of Ki67 mRNA between coated and uncoated iAs (Figure 1F and Appendix A). However, Ki67 expression in NPC was much higher (15%) when compared to iAs-coated (2.1%) or non-coated (2.8%) (Figure 1E). Similarly, mRNA levels of ki67 RUCDR-iAs were approximately 94% lower compared to NPC (Figure 1F) and those of Axol iAs were 97% lower compared to Axol-NPCs (Appendix A). These results confirm that NPCs proliferate at a much higher rate than iAs. A similar pattern of Ki67 protein expression was observed in the case of Axol whereby NPCs exhibited at least two-fold higher levels when compared to iAs (Appendix A). Of note, the Ki67 expression in Axol-iAs was substantially higher (approx. five to seven-fold) compared to RUCDR-iAs at a similar age (compare iA bars in Figure 1E and Appendix A). Similarly, Ki67 of Axol-NPCs were at least two-fold higher than RUCDR-NPCs (Appendix A), suggesting individual hiPSC-derived iAs or NPCs can exhibit considerable variation in proliferation rates.

We then evaluated the staining patterns and quantified the expression of classical astrocyte markers GFAP, S100β, and EAAT-1 through immunofluorescent (IF) staining and flow cytometry in iAs cultured with and without Matrigel coating. The IF staining patterns of GFAP, S100β, and EAAT-1 in iAs were similar irrespective of Matrigel condition and across RUCDR and ATCC cell lines (Figure 2A). Specifically, in RUCDR-iAs, the average expression of GFAP was 72% in Matrigel-coated and 61% in uncoated wells; S100β expression was 69% in Matrigel-coated and 57% in uncoated wells. EAAT-1 expression was 58% and 58% in Matrigel-coated and uncoated conditions, respectively (Figure 2B). Similarly, in ATCC-iAs, average expression of GFAP was 69% and 67%, respectively, S100β was 67% and 66%, and EAAT-1 was 56% and 61% in Matrigel-coated and uncoated conditions, respectively (Figure 2C). Although minor differences in the expression of these markers were noted between coated and non-coated conditions, they were statistically insignificant. These data highlight the similarity in the phenotype of iAs whether they are cultured with or without Matrigel. Importantly, this similarity is maintained across different cell lines. Similar observations were made for the expression of phenotypic markers by flow cytometry. GFAP in RUCDR-iAs was 90% and 92%, S100β was 91% and 88%, and MAP-2 was 10% and 14% in Matrigel-coated and uncoated groups, respectively. (Figure 2D,E). In ATCC iAs, GFAP expression was 53% with Matrigel and 60% without. S100β expression was 67% and 71% with and without Matrigel, respectively, and MAP-2 expression was 16% and 11% in those same groups (Figure 2F). In Axol iA, the expression of GFAP was 95% in Matrigel-coated and 93% in the uncoated group. S100β expression was 66% and 69% in Matrigel-coated and uncoated groups, respectively. MAP-2 expression was 5% with Matrigel coating and 16% without (Figure 2G).

### 2.1. iPSC, NPCs, and iNeurons Require Matrigel Coating for Survival, Unlike iAs

Given that Matrigel is dispensable for iAs survival and proliferation, we assessed whether other CNS cell types could survive without matrix coating. iPSC colonies cultured on Matrigel coating attached and became densely confluent colonies with tight borders at around day 3 (Appendix A) and were positive for OCT-4 (>95%) and TRA-1-60 (>90%) expression (Appendix A). However, iPSC colonies cultured directly on the tissue culture surface without Matrigel coating did not attach or survive. They were seen floating in the culture medium within 24–72 h of plating (Appendix A). We observed an average of five colonies/cm^2^ in Matrigel-coated wells and no colonies in non-coated wells (Appendix A). Similarly, NPCs generated from RUCDR, ATCC, and Axol lines were cultured side-by-side with and without Matrigel coating. In all cases, NPCs cultured on Matrigel coating were densely confluent, viable, able to proliferate (Appendix A), and were positive for Nestin (>80%) and PAX6 (>50%) (Appendix A). Conversely, NPCs cultured without Matrigel did not attach to the surface and were seen floating in the wells the following day (Appendix A). Typical morphological features of neurons (numerous dendrites, prolonged axons, and “nesting”) were observed in RUCDR-iNs and Axol-iNs cultured on Matrigel-coated surfaces (Appendix A); Matrigel-coated neurons cultured without Matrigel did not attach and were seen floating in the medium within 24 h of plating (Appendix A). Flow analyses confirmed the expression of classical neuronal markers, TUJ-1 (>50%) and MAP-2 (>75%), in iNs cultured on Matrigel (Appendix A).

### 2.2. Expression of Key Astrocyte-Specific Genes Are Highly Similar in iAs Cultured with or without Matrigel Coating

GFAP, AQP4, S100β, ACSBG1, and VIM are astrocytic markers, either alone or in combination [45,46,47,48,49,50]. We evaluated the transcription of astrocytic genes and found them to be significantly induced in iAs compared to NPCs in RUCDR (Figure 3A–E). We observed similar induction levels of these genes in Axol cell lines (Figure 3H–L). Further, transcription levels of MAP-2, a mature neuronal-specific marker, were significantly reduced in iAs compared to NPCs (Figure 3F,M). APOE (ε4 isoform) is a primary risk factor for the onset of Alzheimer’s disease (AD). Its protein product acts by disrupting lipid homeostasis in astrocytes and microglia. It was well expressed in NPCs and iAs, as indicated by their low Ct values [51,52,53] (Figure 3G,N). Importantly, there was no significant difference in the expression of all these genes when compared between iAs cultured with and without Matrigel (Figure 3). These results demonstrate that iAs generated without Matrigel coating are appropriately differentiated from NPCs, they are highly enriched, devoid of neuronal contamination, and as such, Matrigel coating is indispensable for their culture and propagation.

### 2.3. Glutamate Uptake and Cytokine Stimulation Are Similar in iAs Cultured with or without Matrigel

Glutamate uptake and clearance from synaptic cleft is a prototypical function of astrocytes to prevent glutamate excitotoxicity and neuronal damage [11,15,54]. We assessed the glutamate uptake ability of iAs cultured with and without Matrigel coating. The percent glutamate uptake in RUCDR iAs increased with time. The pattern exhibited high similarity between coated and uncoated conditions, with a maximum uptake of approximately 20% by 2 h. The percent glutamate uptake in RUCDR iAs from coated wells at 0, 30, 60, and 120 min was not significantly different from that in uncoated wells (Figure 4A). Interestingly, NPCs also took up glutamate, and exhibited a similar increasing pattern of uptake with time; however, their uptake ability was much higher than iAs, at around 51% by 2 h (Figure 4A). This higher uptake level could be simply because of the presence of more cells in the plate. Because our glutamate assay is novel and somewhat indirect, we wanted to ensure that the uptake is real and not occurring due to loss of glutamate due to incubation or degradation by extracellular factors. We used TFB-TBOA, a competitive inhibitor of the glutamate transporters EAAT-1 and EAAT-2. In RUCDR iAs treated with a single dose of DMSO vehicle, a single dose of 10× (1 μmoL), a single dose of 1× (0.1 μmoL), or a continuous dose of 1× TFB-TBOA, there was a 60%, 39%, and 93% reduction in glutamate uptake, respectively (Figure 4B). These data demonstrated that glutamate uptake assay is highly specific and the continuous presence of TFB-TBOA is required to thoroughly block the ability of iAs to take up glutamate via EAAT transporters owing to its competitive inhibitory nature. Further, we also evaluated the ability of iAs to uptake glutamate in relation to their age. On days 3, 9, and 14 of culture in coated and uncoated RUCDR iAs, glutamate uptake was comparable (Figure 4C). These results suggest that as the cells age, their ability to uptake glutamate reduces drastically. Finally, we show that Axol-iAs also exhibit glutamate uptake, approximately 40.1% in Matrigel-coated and 35.1% in non-Matrigel-coated wells (Figure 4D).

Next, we evaluated the transcription of EAAT-1, EAAT-2, and EAAT-3, key genes required for the transport of glutamate in iAs. Although expression of all three genes was detected, EAAT-1 was expressed at higher levels (>100 folds) than EAAT-2 and EAAT-3 (Figure 4E). Interestingly, the fold change in transcription of EAAT-1 in RUCDR iAs cultured without coating was significantly higher than with Matrigel coating (*p* = 0.007). This difference in expression could account for the difference in glutamate uptake between coated and uncoated iAs represented in Figure 4A (19.8% vs. 23.3%, *p* = 0.6). The fold change in transcription for EAAT-2 and EAAT-3 was unchanged between coated and uncoated wells (Figure 4E).

Activated astrocytes are the major source of IL-6 during CNS injury and inflammation. In vitro, many stimuli, including IL1-β and TNF-α, can upregulate IL-6 by activating astrocytes [55]. Stimulation with IL1β and TNFα exhibited approximately >80-fold and >30-fold induction of IL-6 mRNA in RUCDR-iAs. However, IFNγ did not stimulate IL-6 transcription (Figure 4F). Lastly, there was no significant difference in IL-6 transcription in RUCDR iAs with and without Matrigel coating, irrespective of the cytokine used to stimulate IL-6 (Figure 4F). These experiments demonstrate the ability of iAs to be activated by cytokine stimulation and their magnitude of cytokine induction is comparable irrespective of Matrigel presence.

### 2.4. Wnt/β-Catenin Pathway Is Robustly Expressed in iAs Cultured with or without Matrigel

Wnt/β-catenin, an important pathway for cell survival and proliferation, is robustly expressed in astrocytes including HFAs and gliomas, and its disruption mediates astrocytic inflammation and senescence [56,57,58,59,60,61]. We evaluated whether this pathway is robustly expressed in iAs. Β-catenin, a central mediator of this pathway, was highly expressed in iAs both in its total (TBC) and active (ABC) forms, and its expression pattern was similar irrespective of Matrigel coating. The median density of TBC in RUCDR iAs was not statistically different in Matrigel-coated and non-Matrigel-coated, respectively. For ABC, the median density was equivalent in Matrigel-coated and uncoated iAs, respectively (Figure 4G,H). Next, we evaluated the transcription of β-catenin, TCFs, and LEF-1 with qRT-PCR. Transcripts of β-catenin and all three TCFs/LEF-1 were robustly detected in iAs. There was no statistically significant difference in the transcription of TCFs, LEF-1, and β-catenin between Matrigel-coated and uncoated conditions; however, we noted the TCF-4 transcription to be higher in the non-Matrigel-coated group (Figure 4I). We further evaluated whether the pathway is active by transducing the reporter plasmid, TOPFlash in RUCDR-iAs. The pathway was highly active in basal conditions with an average RLU/μg of protein of 19,000–20,000 under DMSO vehicle treatment in coated and uncoated conditions. When these cells were treated with the GSK3 inhibitor, CHIR, the pathway activity was induced approximately 10-fold, with an average RLU/μg of 175,000 to 225,000 (Figure 4J). Further, similar results were observed in Axol-iAs (Figure 4K). The average luminescence in Matrigel-coated and uncoated Axol-iAs treated with a DMSO vehicle was no different between Matrigel conditions. When these cells were treated with CHIR, there was an approximately 14–16-fold increase in luminescence in both groups without statistical differences noted between Matrigel conditions.

### 2.5. The Mitochondrial Respiration Is Indistinguishable between iAs Cultured on or without Matrigel

We assessed oxygen consumption rate (OCR), a marker of mitochondrial efficiency, and the extracellular acidification rate (ECAR), a marker of glycolytic upregulation, in iAs cultured with Matrigel coating or without using Seahorse mito stress assay. Along with OCR and ECAR, the assay measures maximal respiration, basal respiration, ATP-linked respiration, proton (H+) leak, spare respiratory capacity, and non-mitochondrial respiration using various cell-based respiration modulators. In RUCDR iAs, there were no major differences in the OCR between Matrigel-coated and uncoated conditions. Specifically, there is no difference in ECAR, basal mitochondrial respiration, maximal respiration, ATP production, non-mitochondrial respiration, spare respiratory capacity, and mitochondrial coupling efficiency between Matrigel-coated and non-coated iAs (Figure 5A–E,G–I). Proton leak was statistically higher in coated than uncoated iAs (Figure 5F). An increased proton leak typically signifies lower mitochondrial efficiency that translates into increased energy consumption and energy dissipation as heat. At a higher ECAR (Figure 5B), Matrigel-coated cells appear to be more metabolically active at baseline (Figure 5C,E–G) but unable to cope with additional metabolic challenges (Figure 5D,H). The latter finding would suggest a metabolically-stressed cell; however, the coupling efficiency not being affected is unusual (Figure 5I). More work needs to be done to elucidate this issue.

In Axol-iAs, there were no differences observed between Matrigel conditions in OCR at each time point (Appendix A). ECAR was higher in Matrigel-coated iAs at specific time points, although there were no statistical differences (Appendix A). Basal respiration was higher in coated Axol-iAs as measured in OCR (pmol/min/μg of protein), although this difference did not reach statistical significance (Appendix A). Maximal respiration was significantly lower in non-Matrigel-coated iAs (*p* = 0.01) (Appendix A). ATP production was higher in Matrigel-coated iAs; however, this was not statistically significant (Appendix A). Proton leak was higher in Matrigel-coated iAs without reaching statistical significance (Appendix A). Non-mitochondrial O2 consumption was higher in Matrigel-coated iAs without statistical significance (Appendix A). Spare respiratory capacity was significantly reduced in non-Matrigel coated iAs (*p* = 0.03) (Appendix A). There was no difference in coupling efficiency between coated and non-coated iAs (Appendix A). At a lower OCR (Appendix A), non-Matrigel-coated iAs seem to have a reduced capacity for maximal O_2_ utilization (Appendix A) and spare capacity (Appendix A). Whether this is due to true biological differences between these cells or intrinsic variability in the assay is not clear.

### 2.6. The Global Gene Expression Profiles of iAs Cultured with and without Matrigel Are Substantially Similar

We performed RNA sequencing (RNA-seq) followed by exome analysis to assess global gene expression profiles of iAs cultured with or without Matrigel coating. The transcription profiles of iAs showed substantial similarity when cultured with or without Matrigel. The PCA plot showed clustering of samples in proximity on a narrow x and y axis scales (Figure 6A). In RUCDR-iAs, most enriched genes (CPM > 5) were shared between Matrigel coated and non-coated culture conditions as depicted in a Venn diagram (95.97%, Figure 6B). The volcano and scatter plots exhibited tight distribution of gene expression profiles in both RUCDR (Figure 6C,D) and Axol-iAs (Figure 6E,F), demonstrating the virtually identical nature of the exome expression of these cells irrespective of Matrigel condition. There were very few (<5) differentially expressed genes (DEGs) between iAs cultured with or without Matrigel both in RUCDR-iAs and Axol-iAs (volcano plots). The four DEGs found in RUCDR-iAs were *TEX15*, *STMN2*, *GINS2*, and *LIME1*. All were found to be involved in early development (Table 1). Similarly, in Axol-iAs, the only four DEGs identified were *LIN28A*, *LIN28B*, *TRIM71,* and *CECR2* (Table 2), and all were involved in the cellular transformation of embryonic stem cells and other early development processes.

## 3. Discussion

We investigated whether Matrigel coating is necessary for generating phenotypically and functionally active astrocytes (iAs) derived from hiPSCs and NPCs. We compared the morphology, viability, proliferation, and phenotypes of iAs cultured without Matrigel to those cultured on Matrigel.

We show that the morphology of iAs cultured without Matrigel highly resembles that of iAs cultured on Matrigel. Both conditions exhibited stellate-shaped cells with long cellular processes and low overlap between processes. The latter finding suggests that Matrigel coating is not essential for supporting the morphology of iAs. Further, viability and proliferation assays demonstrated no significant differences between iAs cultured with and without Matrigel. The viability of iAs was comparable in both conditions and the proliferation rates, as assessed by Ki67 expression, were also similar. These findings suggest that iAs can maintain their viability and proliferative capacity without Matrigel coating. The expression of astrocytic markers, including GFAP, S100β, and EAAT-1, was evaluated through immunofluorescent staining and flow cytometry. The results indicated that the expression patterns of these markers were similar in iAs cultured with and without Matrigel, regardless of the IPSC-astrocyte line used. The latter demonstrates that the phenotypes of iAs remain consistent irrespective of the presence of Matrigel. Transcriptional analysis of key astrocytic and neuronal genes further supported the similarity between iAs cultured with and without Matrigel. The expression levels of GFAP, AQP-4, S100β, MAP-2, ACSBG1, VIM, and ApoE were not significantly different between the two conditions, indicating that the genotypic transcription of iAs remains comparable, irrespective of Matrigel coating. Furthermore, by investigating the glutamate uptake capacity of iAs, we demonstrated that Matrigel-coated iAs exhibited a similar ability to uptake glutamate through EAAT transporters compared to Matrigel-uncoated iAs. The transcriptional analysis of EAAT-1, EAAT-2, and EAAT-3 genes showed robust transcription of EAAT-1 in Matrigel-coated and Matrigel-uncoated iAs, with significantly higher transcription in Matrigel-coated iAs. At the low Ct values seen, this difference does not seem to have biological significance, particularly in the context of similar protein expression levels and rates of glutamate uptake. Furthermore, we found no significant differences in EAAT-1 transcription during the enriched pathway analysis and RNASeq. EAAT-2 and EAAT-3 transcription were comparable. We also found that the Wnt/β-catenin pathway is robustly expressed in these cells and did not note any difference in its expression irrespective of Matrigel coating. When treated with pro-inflammatory cytokines, iAs upregulated IL-6 transcription proportionally, and no differences were observed relative to Matrigel coating. We also assessed the mitochondrial respiration of these cells. We found that except for proton leak, there were no significant differences in any of the parameters of the Seahorse assay in iAs based on the Matrigel condition.

Some of our unique findings include (1) significantly lower Ki67 transcription and expression in iAs relative to NPCs. This finding indicates the mature nature of our iAs and their low proliferative capacity when compared to progenitor cells such as NPCs [31,62]. (2) Low expression of MAP-2 relative to high expression of GFAP and S100β. This indicates a highly purified astrocyte population with little potential for neuronal contamination. (3) High expression of VIM along with GFAP and S100β. We believe this indicates an iA phenotype corresponding to white matter more than gray matter. GFAP, Vimentin, and Nestin are highly expressed in white matter astrocytes compared to gray matter, and so is EAAT-1. This is following the metabolic support necessary for oligodendrocytes in white matter; these cells can develop membranes up to 100 times the weight of their cell bodies, thus resulting in extremely high metabolic rates and the generation of waste products [63,64,65,66]. Astrocytes are the only cell type capable of handling large loads of glutamate, metabolizing ammonia and maintaining metabolic homeostasis in this compartment [11,14,15,67]. (4) Noticeable induction of EAAT-1 relative to EAAT-2 and EAAT-3 and specific inhibition of glutamate uptake by TFB-TBOA, which is specific to EAAT-1 and EAAT-2. This finding is another indicator of the robust maturity of our iAs given the specific nature of the transporter EAAT-1. Other investigators have applied a more complex methodology in assessing glutamate uptake, including colorimetric measurement of glutamate present in cell lysate and fluorescence emission of intracellular glutamate. We adapted a glutamate uptake assay that utilizes the ability of cells to absorb soluble glutamate from surroundings such as cell media and measured residual glutamate in cell supernatant as our target. Applying an EAAT-1-specific inhibitor also confirms that this transporter is the principal mechanism by which iAs uptake glutamate. (5) Highlighting the importance of studying mitochondrial respiration and metabolism of iAs. Astrocytes are the epicenter of the brain’s TCA cycle, accounting for up to 14% of the brain’s energetic expenditure, more than any other metabolic process [14]. Astrocytes must continually generate a very high rate of ATP to maintain the TCA cycle, and this is generated almost entirely by the electron transport chain through mitochondrial respiration. Thus, studying mitochondrial function in astrocytes is very important for research involving these cells. (6) Importance of studying the Wnt/β-catenin pathway in iAs. We show that this pathway is robustly expressed in iAs. Previous work of our group has shown the importance of this pathway in astrocyte health (glutamate uptake) and disease (HIV latency, Zika entry) alike [56,68,69].

A vial of 10 mL Matrigel costs approximately USD 345, and up to 60–120 μL will be required to coat a single 12-well plate. Thawing, handling, and incubating the plates or other coated surfaces will require at least 90–120 min of processing time. This all adds to any research lab’s costs and labor time, but particularly impacts smaller labs or those in resource-limited settings. Our study shows that Matrigel is dispensable when culturing iAs. This signifies a step forward in optimizing iPSC-based protocols and reducing research costs and time consumption.

Overall, the findings of this study demonstrated that Matrigel coating is not necessary for maintaining the morphology, viability, proliferation, phenotypes, and functionality (glutamate uptake, IL-6 production, robust Wnt/β-cat activity, and mitochondrial respiration) of iAs derived from iPSC and NPCs. These results are important for developing more simplified and cost-effective culture protocols for generating functional astrocytes in vitro. These could benefit various applications in neuroscience research and regenerative medicine.

## 4. Experimental Procedures

### 4.1. Materials Availability

There are restrictions to the availability of iAs and iNs generated in this study due to the lack of an external centralized repository for its distribution and our need to maintain the stock. We are glad to share iAs and iNs with reasonable compensation by request or for their processing and shipping.

### 4.2. Data Availability

All datasets will be publicly accessible in a data repository or shared by the lead contact upon request.

### 4.3. Maintenance and Propagation of iPSCs

hiPSC lines were obtained from Rutgers University Cell and DNA Repository (RUCDR) Infinite Biologics (ID# NN0003920, source: fibroblast, donor: 64-year-old, male. Total passage number: unknown + 4, reprogramming method-retroviral transduction; Sendai persistence and episome persistence not applicable) and American Type Culture Collection (ATCC) (ID#ACS-1031, Manassas, VA, USA, source: bone marrow-derived CD34^+^ T-cells, donor: 27-year-old, Asian, female. Reprogramming method-Sendai viral transduction; episome persistence not applicable) maintained in complete mTeSR+ medium (StemCell Technologies; Vancouver, BC, Canada) in the presence of 0.1% penicillin/streptomycin and propagated as small aggregates using ReLeSR (StemCell Technologies). All iPSC colonies were cultured at 37 °C in a 5% CO_2_ on Matrigel-coated tissue culture plates. Coating of plates was performed according to the manufacturer’s instructions (Corning; Corning, NY, USA). Briefly, aliquots of Matrigel were thawed on ice, diluted 100-fold in ice-cold DMEM/F-12 culture medium (Fisher Scientific; Hampton, NH, USA), and vortex mixed. Appropriate volumes per well (1 mL, 500 μL, 250 μL or 125 μL to 6-, 12-, 24- or 96-well plates, respectively) were added and incubated for 1 h at room temperature or 37 °C in 5% CO_2_. iPSCs were routinely validated for markers OCT4 (StemCell: 60093PE.1) and TR-1-60 (StemCell: 60064PE.1) using immuno-fluorescent labeling and flow cytometry.

### 4.4. Differentiation of iPSC into Astrocytes and Neurons via NPC

iPSCs were differentiated to NPC by a neural induction medium containing dual SMAD inhibitors in the presence of 0.1% penicillin/streptomycin (cNIM-SMADi, StemCell Technologies) through monolayer culture protocol (StemCell Technologies). Briefly, iPSCs were dissociated into single cells with accutase (Stem Cell Technologies) and plated at 2 × 10^6^ cells per well of a 6-well plate on day 1 in mTeSR+. The next day (day 0), cells were replaced with fresh cNIM-SMADi and replaced with this fresh media every day until NPC differentiation was completed. NPCs were expanded for at least three passages (7–10 days per passage) before differentiating into astrocytes. A commercially available human NPC line was obtained from Axol (ax0019, donor: female, Cambridge, UK) and maintained on cNIM-SMADi media on Matrigel-coated plates. Using flow cytometry, NPCs were routinely validated for their specific markers, Pax-6 (Abcam: ab5790) and Nestin (Abcam: ab22035).

NPCs were differentiated into astrocytes as described [31]. Briefly, NPCs were dissociated to single cells by accutase and seeded at low density (15,000 cells/cm^2^) on Matrigel-coated plates. On day 1, cells were incubated in cNIM-SMADi and on day 0, the medium was replaced with fresh complete astrocyte medium (cAM) (ScienCell Research Laboratories. Carlsbad, CA, USA: complete kit) containing 2% FBS, astrocyte growth supplement (AGS, Cat. No. 1852), and penicillin-streptomycin solution (P/S, Cat. No. 0503). From day 2, cells were fed every other day for 30 days with fresh cAM. When the cells reached 90–95% confluency (~every 8–10 days), they were dissociated into single cells with accutase, split to the initial low seeding density in cAM, and cultured. After 60 days of differentiation, astrocytes were either used for experiments or frozen for later use. When cultured beyond day 110, we noticed a deterioration in the functionality of iAs concerning glutamate uptake and downregulation in the expression of phenotypic markers such as GFAP, S100β, and AQP-4.

A mixed population of forebrain-type primary neurons (glutamatergic and gabaergic) was generated from RUCDR-NPCs using forebrain neural differentiation followed by maturation kits (StemCell Technologies) according to the manufacturer’s instructions. Neurons were validated for their specific markers, MAP-2 (intracellular, neuronal marker) (clone AP20, Cat #sc-32791, Santa Cruz Biotechnology, Dallas, TX, USA) and anti-tubulin β3 (TUBB3)/AF647 (intracellular, neuronal marker) (clone TUJ-1, Cat #801210, BioLegend, San Diego, CA, USA) using flow cytometry.

### 4.5. Morphology and Cell Quantitation

Images of iAs in Matrigel-coated and non-Matrigel coated plates were captured by bright field microscopy using a Keyence BZ-810 fluorescence microscope (Keyence, IL, USA) at 200× magnification. For each well, images from four different high-power fields were obtained. We assessed their morphology by direct inspection and evaluated for typical morphological features of iAs (stellate or fusiform shape, elongated processes, low overlap between processes of different cells). For cell counts, cells were gently washed with sterile PBS, detached with accutase, resuspended in DMEMF12, stained with trypan blue, and counted using a TC20™ automated cell counter (Bio-Rad, Hercules, CA, USA). Three separate wells were counted as replicates for each condition and time point.

### 4.6. Immunostaining, Histochemistry, and Imaging

Immunostaining was performed as previously described [70]. Briefly, 50,000 cells were seeded on a 24-well glass bottom plate (In Vitro Scientific, Mountain View, CA, USA, #P24-0-N) and cultured for 24 h. Cells were fixed with 400 μL PFA (4%) and washed thrice for 5 min in PBS. To permeabilize, 400 μL of 0.1% triton X-100 in PBS was added for 15 min at RT. Cells were washed three times for 5 min in PBS. Fixed and permeabilized cells were blocked for 1 h with 500 μL of blocking buffer (5% normal donkey serum, 2% BSA, and 0.25% triton X-100 in PBS). Primary antibodies were added (1:500) in 1:1 diluted blocking buffer: EAAT1 in mouse (Santa Cruz Biotechnology, #SC-515839), S100β in rabbit (Novus Biologicals, Centennial, CO, USA, #NBP1-87102), and GFAP in rat (Thermo Fisher Scientific, Waltham, MA, USA, #13-0300) and incubated at 4 °C overnight. Cells were washed three times for 5 min in PBS. Fluorescent conjugated secondary antibodies were incubated (1:500) with corresponding primary antibody species for 45 min at RT: donkey anti-mouse AF-488 (Abcam, #150109), donkey anti-rabbit AF-594 (Abcam, #150072), and donkey anti-rat AF-647 (Abcam, #150155). After incubation, cells were washed three times for 5 min in PBS and allowed to air dry before applying DAPI counterstain (Thermo Fisher Scientific). Imaging was performed with a Keyence BZ-810 fluorescence microscope using the BZ-X800 viewer imaging software (Copyright© 2018 Keyence Corporation, Itasca, IL, USA).

### 4.7. Flow Cytometry

Cells cultured on Matrigel-coated and non-coated plates were detached with accutase, washed twice with sterile PBS, resuspended in approximately 100 μL of perm/wash buffer, incubated with Fc block (clone Fc1, Cat #564220, BD Biosciences, East Rutherford, NJ, USA) for 10 min as per instructions, and subjected to either cell surface staining for surface markers using perm/wash buffer or intracellular staining using a perm/fix protocol (BD Biosciences). The following primary conjugated antibodies for phenotypic markers were used for staining astrocyte markers: GFAP/BV421 (intracellular, clone 2E1.E9, cat #644710, BioLegend, San Diego, CA, USA), S100β/FITC (intracellular, astrocyte marker) (clone s100B/1012, Cat #NBP2-59619F, Novus Biologicals); neuronal, MAP-2/PE (intracellular, clone AP20, Cat #sc-32791, Santa Cruz Biotechnology) and tubulin β3 (TUBB3)/AF647 (intracellular, clone TUJ-1, Cat #801210, BioLegend); proliferation marker, Ki-67/PE (intracellular, clone B56, Cat #556027, BD Biosciences). Cells were incubated with antibodies at 4 °C for 60 min in the dark, washed thoroughly with perm/wash buffer, resuspended in perm/wash buffer, and run on a Fortessa flow cytometer (BD Biosciences). Data were analyzed on FlowJo™ v10.8 Software (BD Life Sciences) and normalized to the respective IgGs. In the case of iAs, data were represented as percent positive events compared to Matrigel-coated samples.

### 4.8. Quantitative RT-PCR

RNA extraction was performed utilizing the RNeasy^®^ mini kit (Qiagen, Germantown, MD, USA), per the manufacturer’s instructions. DNAse digestion was performed in the column for 15 min at RT. The quality and quantity of RNA were verified with a NanodropTM 2000/2000c spectrophotometer (Fisher Scientific). All samples had 260/280 ratios > 1.8 and 260/230 ratios between 2.0 and 2.2 with adequate melt curve shapes. Complementary DNA (cDNA) was synthesized using a Qscript cDNA synthesis kit (QuantaBio, Beverly, MA, USA) per the manufacturer’s instructions. Samples were incubated in a T-100 thermal cycler (Bio-Rad).

Real-time quantitative PCR (qPCR) was performed on cDNA samples (at least 1/20 volume) using SYBR green master mix. Samples were loaded on a 96-well fast block and run on a QuantStudio 7 Flex (Thermo Fisher) instrument. Protocol settings were as follows: hold stage (95 °C for 10 min), 45 cycles of PCR stage (95 °C for 20 s and 60 °C for 1 min), and melt curve stage (95 °C for 15 s and 60 °C for 1 min). Primers used are listed below. Cycle threshold (CT) values were normalized to a housekeeping gene (GAPDH) and expressed as fold changes compared to either the Matrigel-coated group or parent NPC.

### 4.9. Primers

EAAT1 F(5′-GGTTGCAAGCACTCATCAC-3′), R(5′-CACGCCATTGTTCTCTTCCAGG-3′); EAAT2 F(5′-TGCCAACAGAGGACATCAGCC T-3′), R(5′-CAGCTCAGACTTGGAGAGGTGA-3′); EAAT3 F(5′-CGAAAGAACCCTTTCCGATTTGC-3′), R(GAAGGTGACAGGCAGTGTTGCT-3′); F(5′-CTGCCATACCAACTTCAGCACC-3′), R(5′-ATAGGCACGGATGTGGTACTGG-3′); Ki67 F(5′-TAACACCATCAGCAGGGAAAG-3′), R(5′-CTGCACTGGAGTTCCCATAAA-3′); GFAP F(5′-CTGGAGAGGAAGATTGAGTCGC-3′), R(5′-ACGTCAAGCTCCACATGGACCT-3′); VIM F(5′-AGGCAAAGCAGGAGTCCACTGA-3′), R(5′-ATCTGGCGTTCCAGGGACTCAT-3′); S100β F(5′-GAAGAAATCCGAACTGAAGGAGC-3′), R(5′-TCCTGGAAGTCACATTCGCCGT-3′); MAP-2 F(5′-AGGCTGTAGCAGTCCTGAAAGG-3′), R(5′-CTTCCTCCACTGTGACAGTCTG-3′); *ACSBG1* F(5′-CCCCTTGACCTGTGATGACC-3′), R(5′-GAGACGGGATGGACTTGGA-3′); *APOE* F(5′-GAGCAGGCCCAGCAGATAC-3′), R(5′-CTGCATGTCTTCCACCAGGG-3′); GAPDH F(5′-GTCTCCTCTGACTTCAACAGCG-3′), R(5′-ACCACCCTGTTGCTGTAGCCAA-3′); IL-6 F(5′-AGACAGCCACTCACCTCTTCAG-3′), R(5′-TTCTGCCAGTGCCTCTTTGCTG-3′); Beta-catenin F(5′-CACAAGCAGAGTGCTGAAGGTG-3′), R(5′-GATTCCTGAGAGTCCAAAGACAG-3′); LEF1 F(5′-CTACCCATCCTCACTGTCAGTC-3′), R(5′-GGATGTTCCTGTTTGACCTGAGG-3′); TCF3 F(5′-CCAGACCAAACTGCTCATCCTG-3′), R(5′-TCGCCGTTTCAAACAGGCTGCT-3′); TCF4 F(5′-GCCTCTTCACAGTAGTGCCATG-3′), R(5′-GCTGGTTTGGAGGAAGGATAGC-3′); AQP-4 F(5′-GCCATCATTGGAGCAGGAATCC-3′), R(5′-ACTCAACCAGGAGACCATGACC-3′); TCF1 F(5′-CTGACCTCTCTGGCTTCTACTC-3′), R(5′-CAGAACCTAGCATCAAGGATGGG-3′).

### 4.10. Glutamate Uptake Assay

Glutamate uptake was measured as described with slight modifications [56]. Briefly, astrocytes were washed gently and incubated for 10 min at RT with calcium-free HBSS. Cells were then incubated with glutamate at a concentration of 50 μM in calcium-containing HBSS for 2 h at 37 °C in a 5% CO_2_. A cell-free well containing the same volume of glutamate containing HBSS was incubated in parallel as a reference point. Following the incubation period, the supernatants were collected from all wells, spun at 5000 rpm for 5 min to remove debris, and stored at −20 °C till further use. The concentration of glutamate in the HBSS-Ca+ buffer was estimated using PicoProbe^TM^ fluorometric glutamate assay kit (BioVision, Milpitas, CA, USA) as per instructions, and supernatants were loaded into a 96-well microplate that was pre-loaded with a proprietary assay buffer. HBSS-Ca+ buffer was used for the background signal, and a standard curve for glutamate concentrations was generated as described in the protocol. Optical density was measured in a Cytation 3 (BioTek, Winooski, VT, USA) microplate reader at 450 nm. Differences in the glutamate concentration between reference points and samples were calculated and represented as percent glutamate uptake. We also used a specific glutamate uptake inhibitor TFB-TBOA (Tocris Bioscience, Bristol, UK), either continuously incubated with the cells throughout the assay or preincubated for 2 h before adding glutamate to cells.

### 4.11. Seahorse Cell Mito Stress Test Assay

The Seahorse cell mito stress test was performed as previously described [71]. Briefly, astrocytes were seeded in 24-well seahorse plates at 5 × 10^4^ cells per well in cAM. Immediately before the assay, media were changed to DMEM-XF supplemented with 1 mM pyruvate, 2 mM glutamine, and 10 mM glucose at a pH of 7.4. Cells were then incubated in a 37 °C non-CO_2_ incubator for 1 h. The mito stress test is a plate-based live-cell assay. The instrument uses built-in injection ports on XF sensor cartridges to add modulators of respiration into the wells, thus generating key information on mitochondrial function. Each well was injected with the key modulators oligomycin (2.5 µM), FCCP (2 µM), and rotenone/antimycin A (0.5 µM). Basal respiration, ATP production, proton leak, maximal respiration, spare respiratory capacity, and non-mitochondrial respiration were measured per a previously published method [71,72]. The assay was run in a Seahorse XFe24 analyzer (Agilent Technologies, Santa Clara, CA, USA). Following the assay, cells were lysed, and the protein content was determined by the BCA assay (Thermo Fisher Scientific). All values from the Mito stress assay were normalized to the protein content per well. Data were analyzed on Google Sheets and GraphPad Prism5.

### 4.12. TOP-Flash Reporter Assay and Western Blotting

iAstrocytes from both RUCDR and Axol were split and transduced with a lentiviral particle containing 7TFP (#24308, Addgene, Watertown, MA, USA) as the transfer plasmid. Expression vector 7FTP harbors a firefly luciferase gene downstream of a minimal promoter containing 7× TCF/LEF consensus binding elements and is termed as TOPflash reporter. Second-generation lentiviral (LV) particles were produced in HEK293 cells using three plasmids, pMD2.G (#12259, Addgene), psPAX2 (#12260, Addgene), and 7TFP. Briefly, high quality, endotoxin frendotoxin-freetained from maxi prep kit (Qiagen) were transfected at a ratio of (3(TTFP):2 (psPAx2):1 (pMD2.G) using calcium phosphate mediated transfection (Millipore Sigma, Burlington, MA, USA). One day later, cells were gently washed and replaced with fresh, complete DMEM. Supernatants containing viral particles were collected at 48 and 72 h post-transfection, spun at 5000 rpm for 10 min to remove cell debris, and subjected to viral concentration using LentiX (Takara Bio, Chicago, IL, USA). LV particles were quantified using a P24 ELISA kit as per instructions (Abcam). iAs were transduced with 5–10 ng/mL of LV particles. Then, 36 h post-transduction, a few wells were treated with CHIR99021 (CHIR, Tocris) for 24 h, an inhibitor of GSK3, which subsequently promoted saturation of β-catenin in the cytoplasm. Cells were lysed using a passive lysis buffer and centrifuged at 5000 rpm for 5 min to remove debris. Luciferase activity was measured using the Nano-Glo^®^ luciferase reporter assay (Promega, Madison, WI, USA) as per instructions. Total protein concentration was measured using a BCA assay. Data were normalized to the protein concentration in the samples and represented as RLU/μg of protein.

Western blotting was performed as previously described [57]. Briefly, 10–15 μg of cell lysate was loaded and run on a 10% SDS-PAGE, transferred to nitrocellulose, and blocked for 1 h at RT in SuperBlock. Overnight incubation with primary antibodies to active β-catenin (mouse, 1:4000) and GAPDH (rabbit, 1:25,000) was performed. Then, it was washed with 1x-TBS-Tween-20 (0.1%) five times for 5 min. Secondary antibody incubation for 2 h at RT (anti-mouse-HRP 1:2000 and anti-rabbit-HRP 1:50,000) was performed. Then, it was washed as before five times for 5 min. The blot was subjected to a 2 min incubation with Femto substrate and a 15 s exposure before developing. Densitometry was analyzed using ImageJ. The total density for each individual band was measured, and an average of two bands is represented in the figure.

### 4.13. RNA Sequencing (RNA-Seq) and Bioinformatics Analysis

Total RNA was purified using the miRNeasy kit (Qiagen) and used for sequencing library construction. RNA samples were checked for purity using NanoDrop and analyzed for integrity using 4200 TapeStation (Agilent). Relative levels of remaining DNA were checked by dual RNA/DNA measurements using a Qubit fluorometer (Invitrogen). DNA amounts did not exceed 10% of the total amount of nucleic acids. Sequencing libraries for Illumina sequencing were prepared using 50 ng of total RNA per sample. Library prep was carried out with the Universal plus mRNA-seq library preparation kit (Tecan/NuGen; #0520-A01), as written in the product manual (NuGen; M01485 v5). In brief, RNA underwent poly (A) selection, enzymatic fragmentation, and generation of double-stranded cDNA using a mixture of oligo (dT) and random priming. The cDNA underwent end repair, ligation of dual-index adaptors, strand selection, and 14 cycles of PCR amplification. The number of cycles was determined by qPCR on a small aliquot of the unamplified libraries. All intermediate purification steps and final library purification were carried out using Agencourt AMPure XP beads (Beckman Coulter, Brea, CA, USA; #A63881). Final amplified libraries were measured with the Qubit 1 double-stranded DNA (dsDNA) HS assay kit (Invitrogen; #Q33231), and fragment size distribution was confirmed to be approximately 470 bp using the D5000 ScreenTape assay (Agilent; #5067-5588 and #5067-5589). The concentration of the final library pool was confirmed by qPCR and subjected to test sequencing to check sequencing efficiencies and adjust the proportions of individual libraries accordingly. The pool was purified with the Agencourt AMPure XP beads (Beckman Coulter; #A63881), quantified by qPCR using KAPA library quantification kit, and sequenced on a NovaSeq 6000 S4 flow cell, 2 150 bp, approximately 30 M clusters per sample, at the University of Illinois Roy J. Carver Biotechnology Center High-Throughput Sequencing and Genotyping Unit (Chicago, IL, USA).

Clustering was performed using k-means clustering, selecting the number of clusters (k) with reproducibility clustering statistics, like the approach outlined previously [73]. K-means clustering was performed with 10 random initializations on a range of cluster numbers k (2 to 20). For each k, the reproducibility of the repeated clustering runs was computed by comparing the pairwise distance between clustering results as the number of co-clustered feature pairs shared between results divided by the number of co-clustered feature pairs within each result individually. This difference was averaged across all result pairs for each value of k, and the largest k with an average distance less than 1 × 10^−5^ was selected as the optimal k with highly reproducible clusters. Enrichment clusters for features that were significant (q, 0.05) were generated using Cytoscape. Venn diagrams were also generated using an open-source Venn diagram viewer.

### 4.14. Statistics

Figures were plotted using GraphPad Prism v5.0. Statistical analysis was performed using Prism (GraphPad) and Google Sheets. All values are represented as mean with SEM. All experiments were performed with a minimum of 3 technical replicates and 3 biological replicates. Variables were compared using a two-tailed, unpaired *t*-test with a *p*-value < 0.05 considered significant. If the *p*-value reaches significance, it is represented with an asterisk (*). Statistical analysis has been performed between coated and uncoated groups for all represented data. The absence of an asterisk denotes a lack of statistical significance.

## Figures and Tables

**Figure 1 cells-12-02357-f001:**
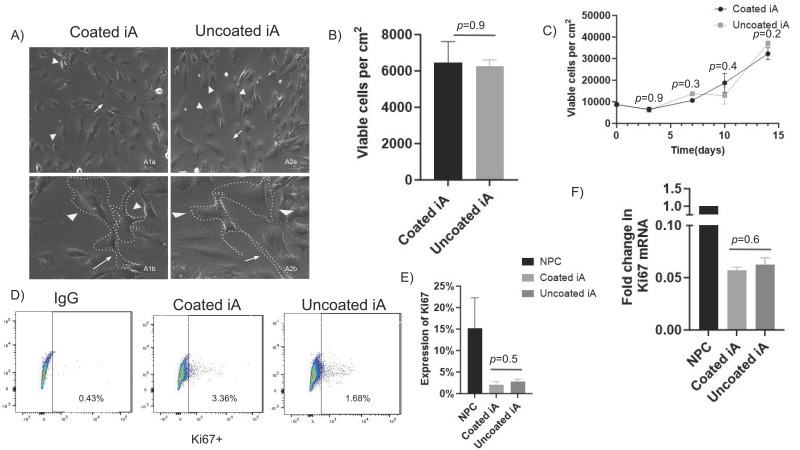
Matrigel coating is not required to maintain morphology, viability, and proliferation of iAs in vitro. RUCDR-iAs were cultured with and without Matrigel coating. (**A**) Phase contrast images (20× magnification) showing typical astrocyte star-like cellular morphology (arrowheads) with elongated cell processes and early network formation (arrows) in Matrigel-coated (A1) and uncoated (A2) iAs. Scale, 50μΜ. Note: A1a is a representative image, and A1b is a cropped-out portion of A1a to show additional detail (Matrigel. Similarly, A2a is a representative image, and A2b is a cropped-out portion of A2a to show additional detail. (**B**) Quantification of the total number of viable cells per cm^2^ of well surface area 72 h post culture. (*n* = 3). (**C**) Proliferation rate of iAs as measured by counting viable cells at various days of culture with and without Matrigel coating (*n* = 3). (**D**) Representative dot plots of Ki67 expression in RUDCR-iAs cultured with and without Matrigel coating. (**E**) Percent expression of Ki67+ cells in iAs plated with and without Matrigel and NPCs plated with Matrigel coating via quantitative flow cytometry (*n* = 3). (**F**) Fold change in Ki67 mRNA replication in RUCDR-iAs plated with and without Matrigel coating compared to NPC (*n* = 3). *p*-values determined by unpaired two-tailed *t*-test.

**Figure 2 cells-12-02357-f002:**
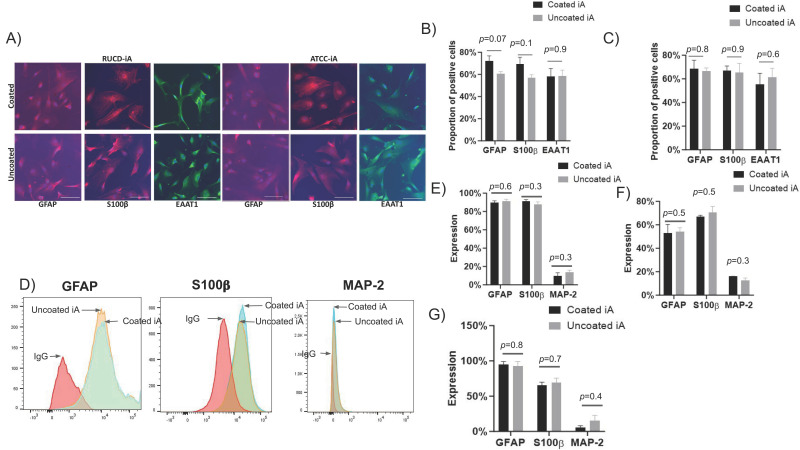
iAs efficiently express classical phenotypic markers in vitro irrespective of Matrigel coating. RUCDR-iAs and ATCC-iAs were cultured on glass or tissue culture plates with and without Matrigel coating for 72 h and subjected to IF or flow cytometry. (**A**) IF staining of RUCDR-iAs and ATCC-iAs showing expression of GFAP, S100β, and EAAT1 when plated with and without Matrigel coating (200× magnification). Scale, 50 μΜ. (**B**) Proportion of cells positive for GFAP, S100β, and EAAT-1 (relative to DAPI staining) in RUCDR-iAs (*n* = 4). (**C**) Proportion of cells positive for GFAP, S100β, and EAAT-1 (relative to DAPI staining) in ATCC-iAs (*n* = 4). (**D**) Representative flow histograms showing GFAP, S100β, and MAP-2 expression in RUCDR-iAs. (**E**–**G**). Quantitative flow for GFAP, S100β, and MAP-2 expression in RUDCR-iAs, ATCC-iAs, and Axol-iAs with and without Matrigel, respectively (*n* = 4). *p*-values determined by unpaired two-tailed *t*-test. Abbreviations: GFAP: glial fibrillary acidic protein, iA: iPSC-derived astrocytes, MAP-2: microtubule-associated protein 2.

**Figure 3 cells-12-02357-f003:**
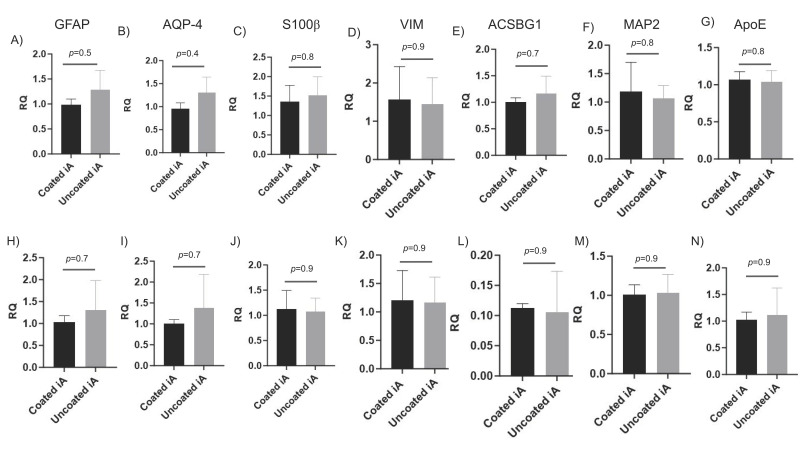
Transcription of astrocyte-specific genes is highly induced in iAs with or without matrigel coating. iAs were cultured side-by-side with or without Matrigel coating. RNA was extracted 72 h later and subjected to qRT-PCR. Quantitative expression of astrocyte-specific genes was normalized to GAPDH mRNA first and then represented as fold change relative to the endogenous marker. Shown are astrocyte-specific markers GFAP (**A**), AQP-4 (**B**), S100β (**C**), VIM (**D**), ACSBG1 (**E**), and neuronal markers MAP-2 (**F**), and ApoE (**G**) in RUCDR cell line (*n* = 6). Similarly, shown are astrocyte-specific markers GFAP (**H**), AQP-4 (**I**), S100β (**J**), VIM (**K**), ACSBG1 (**L**), and neuronal markers MAP-2 (**M**), and ApoE (**N**) in Axol cell line (*n* = 3). *p*-values determined by unpaired two-tailed *t*-test. Abbreviations: ACSBG: acyl-CoA synthetase bubblegum family member 1, ApoE: apolipoprotein E, AQP-4: aquaporin-4, GFAP: glial fibrillary acidic protein, iA: iPSC-derived astrocytes, MAP-2: microtubule associated protein 2, RQ: relative quantity, VIM: vimentin.

**Figure 4 cells-12-02357-f004:**
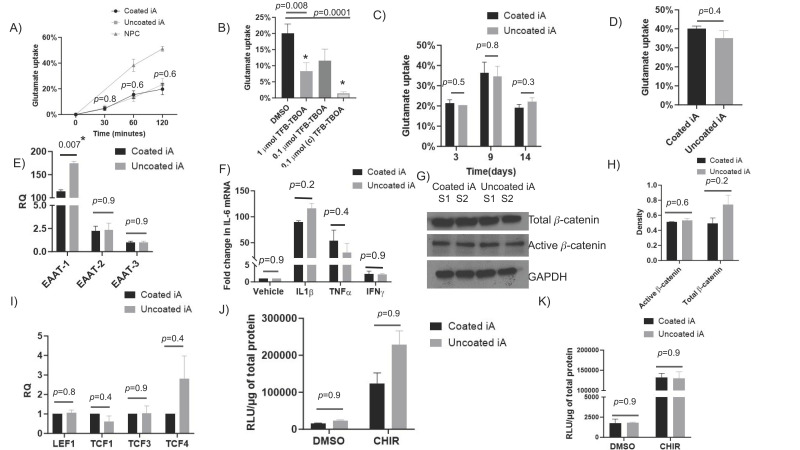
iAs cultured with or without Matrigel exhibited efficient glutamate uptake, activation by cytokine treatment, and Wnt/β-catenin pathway. (**A**) RUCDR-iAs (with and without coating) and NPCs (with coating) were cultured for 72 h, and then percent glutamate uptake was assessed at 30, 60, and 120 min (*n* = 4). (**B**) RUCDR-iAs were cultured for 72 h in Matrigel uncoated wells and pre-treated for 2 h with TFB-TBOA at 1 μM or 0.1 μM and then incubated with 50 μM glutamate or continuously treated with 0.1 μM of TFB-TBOA along with 50 μM glutamate and percent glutamate uptake was determined (*n* = 8). (**C**) RUCDR-iAs were cultured for various time points indicated, and glutamate uptake was assessed and represented as average percent (*n* = 3). (**D**) Axol-iAs (with and without coating) were cultured for 72 h, and then percent glutamate uptake was assessed at 120 min (*n* = 3). (**E**) RUCDR-iAs from Matrigel-coated wells and NPCs from coated wells were lysed and RNA isolated. EAAT1, EAAT2, and EAAT3 transcription was measured by RT-PCR normalized to GAPDH and represented as fold change relative to NPCs (*n* = 3). (**F**) RUDCR-iAs were treated with 10 ng/mL of IL-1β, TNF-α, and INF-γ or vehicle control for 24 h, RNA was extracted, and IL-6 transcription was measured by qRT-PCR (*n* = 3). (**G**) RUCDR-iAs were cultured on Matrigel-coated and uncoated wells for 72 h, protein was extracted, and Western blot was performed for GAPDH, total, and active β-catenin. (**H**) Densitometric quantification of Western blot for total and active beta-catenin normalized to GAPDH (*n* = 2). (**I**) Fold change in mRNA transcription of LEF1, TCF1, TCF3, TCF4, and β-catenin in RUCDR-iAs cultured on Matrigel-coated and uncoated wells. Transcription in coated cells was used as reference (*n* = 3). (**J**,**K**) Luciferase assay to measure TOPflash reporter activity before and after treatment with CHIR, a GSK3b inhibitor in RUCDR-iAs (*n* = 3) and in Axol-iAs (*n* = 4), respectively. *p*-values determined by unpaired two-tailed *t*-test, an asterisk (*) represents *p*-value <0.05. Abbreviations: CHIR: 6-((2-((4-(2,4-dichlorophenyl)-5-(5-methyl-1H-imidazol-2-yl)pyrimidin-2-yl)amino)ethyl)amino)nicotinonitrile hydrochloride, DMSO: dimethyl sulfoxide, GAPDH: glyceraldehyde 3-phosphate dehydrogenase, LEF: lymphoid enhancer factor, RLU: relative light unit, RQ: relative quantity, TCF: T-cell factor, TFB-TBOA: (2S,3S)-3-(3-(4-(trifluoromethyl)benzoylamino)benzyloxy)aspartate.

**Figure 5 cells-12-02357-f005:**
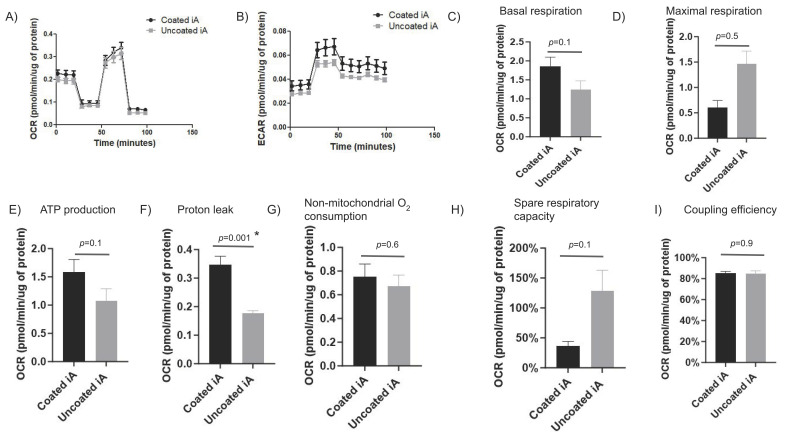
There are only minor differences in mitochondrial respiration between iAs cultured with and without Matrigel. RUCDR-iAs were cultured for 72 h on Matrigel-coated or uncoated condition, seeded in 24-well seahorse plates, and subjected to measurements for mitochondrial function as described in Materials and Methods. (**A**) Oxygen consumption rate (OCR), a marker of overall mitochondrial function, was measured, and values expressed in pmol/min relative to μg of protein per well. (**B**) Extracellular acidification rate (ECR), a marker of glycolytic overdrive and resultant lactic acid generation, was measured and found to be similar irrespective of Matrigel condition. (**C**) Basal respiration (or cellular metabolism at rest) was calculated and found not statistically different between Matrigel conditions. (**D**) Maximal respiration, a measure of peak mitochondrial function under stress, was evaluated and found to be similar between Matrigel-coated and non-coated iAs. © ATP production was similar in non-Matrigel-coated RUCDR iAs compared to Matrigel-coated iAs. (**F**) Proton leak, a measure of mitochondrial membrane integrity, was statistically higher in RUCDR iAs with Matrigel coating than those without coating. (**G**) Non-mitochondrial oxygen consumption, a measure of cytoplasmic glycolysis as a source of ATP, was slightly higher in Matrigel-coated relative to non-coated iAs without a statistical difference. (**H**) Spare respiratory capacity, a measure of peak mitochondrial performance relative to basal conditions, was lower in Matrigel-coated RUCDR-iAs. It was not statistically significant. (**I**) Coupling efficiency, a measure of ATP production capacity relative to basal conditions, was similar in iAs with or without Matrigel coating. All experiments represent three biological replicates. *p*-values determined by unpaired two-tailed *t*-test, an asterisk (*) represents *p*-value < 0.05. Abbreviations: ECAR: extra cellular acidification rate, iA: iPSC-derived Astrocytes, OCR: oxygen consumption rate.

**Figure 6 cells-12-02357-f006:**
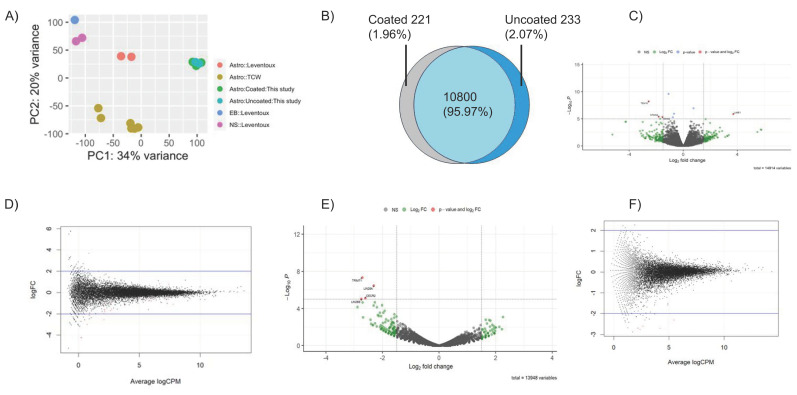
RNAseq analysis exhibited substantial similarity in global gene expression profiles for iAs cultured with or without Matrigel. RUCDR-iAs and Axol-iAs were cultured for 3 days in Matrigel-coated or uncoated condition. Total RNA extracted and subjected to RNAseq analysis as explained in Materials and Methods. (**A**) Principal component analysis (PCA) of samples from this study between Matrigel coated and non-coated samples (RUCDR-iAs and Axol-iAs) and publicly available studies with iPSc-derived astrocytes. (**B**) Venn diagram for highly enriched genes (CPM > 5) showing ~98% overlapping between coated and non-coated conditions in RUCDR-iAs. (**C**) Volcano plot showing the distribution of genes between coated (**left**) and uncoated (**right**) RUCDR-iAs. Green dots indicate a Log_2_ fold change > 2 and −Log_10_P < 5, and red dots (labeled) indicate a Log_2_ fold change > 2 and −Log_10_P > 5. (**D**) Scatter plot showing the distribution of expression profiles of genes in RUCDR-iAs cultured with and without Matrigel. *p*-values determined by unpaired two-tailed *t*-test. (**E**) Volcano plot showing distribution of genes between coated (left) and uncoated (right) Axol-iAs. Green dots indicate a Log_2_ fold change > 2 and −Log_10_P < 5 and red dots (labeled) indicate Log_2_ fold change > 2 and −Log_10_P > 5. (**F**) Scatter plot showing distribution of expression profiles of genes in Axol-iAs cultured with and without Matrigel. Abbreviations: Astro: iPSc-derived Astrocytes, NS: neurosphere, EB: embryoid body.

**Table 1 cells-12-02357-t001:** Differentially expressed genes in RUCDR-iAs cultured with and without Matrigel.

Gene Symbol	Full Name	Function
*TEX15*	Testis expressed 15, meiosis and synapsis associated	DNA double-strand break repair, chromosome synapsis, and meiotic recombination of spermatocytes.
*STMN2*	Stathmin 2	Stathmins participate in microtubule dynamics and signal transduction. The encoded protein plays a regulatory role in neuronal growth
*GINS2*	GINS complex subunit 2	Formation of the complex is essential for DNA replication
*LIME1*	Lck interacting transmembrane adaptor 1	This gene encodes a transmembrane adaptor protein that links the T and B-cell receptor stimulation to downstream signaling pathways via its association with the Src family kinases Lck and Lyn, respectively

**Table 2 cells-12-02357-t002:** Differentially expressed genes in Axol-iAs cultured with and without Matrigel.

Gene Symbol	Full Name	Function
*LIN28A*	Lin-28 homolog A	Posttranscriptional regulator of genes involved in developmental timing and self-renewal in embryonic stem cells
*LIN28B*	Lin-28 homolog B	Highly expressed in testis, fetal liver, placenta, and in primary human tumors, and cancer cell lines. Gene product represses let-7 family of microRNAs and derepresses let-7 targets, which facilitates cellular transformation
*TRIM71*	Tripartite motif containing 71	E3 ubiquitin-protein ligase that binds with miRNAs and maintains the growth and upkeep of embryonic stem cells
*CECR2*	Cat eye syndrome chromosome region	A small supernumerary chromosome is present in this syndrome

## Data Availability

The data presented in this study are openly available at: 10.3390/cells9122680 [74] and 10.1016/j.stemcr.2017.06.018 [31].

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
