# Peer review of "An Efficient and Cost-Effective Approach to Generate Functional Human Inducible Pluripotent Stem Cell-Derived Astrocytes"

_cells, 2023, doi:10.3390/cells12192357_

Round 1
Reviewer 1 Report
To summarize, the authors propose to culture iPSC-derived astrocytes, as well as astrocytes derived from neuronal progenitor cells (NPCs), in the absence of Matrigel substrate or PLO/laminin coating, based on their observations that, unlike neurons or neuronal progenitor cells (NPCs), they do not require Matrigel coating.
In this article, the authors demonstrated that, indeed, morphology, viability, proliferation and phenotypes (expression of classical astrocyte markers, for example GFAP, S100β, and EAAT-1) of such iPSC-derived and NPC-derived astrocytes cultured in absence of Matrigel closely resemble whose cultured on Matrigel. Their response to cytokine treatment, mitochondrial respiration and glutamate uptake were also similar in cells cultured on Matrigel-coated and uncoated surfaces.
Aforementioned approach is cost-effective and therefore of interest to researchers working with induced astrocytes.
Line105. “We cultured iAs generated from two iPSC lines 105 (RUCDR-iAs and ATCC-iAs) and from an NPC line (Axol-iAs)”. As all types of cells were introduced earlier in the text, to avoid redundancy I suggest to write instead “We cultured iAs generated from two iPSC lines and from an NPC line...”
I recommend accepting the manuscript in current form.
Reviewer 2 Report
This is a very well written manuscript describing a thorough experimental design that provides a novel protocol to generate abundant human astrocytes that can be used in neurodegeneration research. This work has the potential to expand studies into astrocyte biology by providing a practical route to generating astrocyte cultures for labs of all size. The background is sufficient. The methods are detailed and provide sufficient information to be readily reproduced. The results are clear and thorough. The Discussion is comprehensive. I have a few minor comments:
- The axes titles are too small to be legible and should be increased
- The age of the iPSC donors are not discussed nor is it discussed whether epigenetic signatures are retained. This is important for age-related research and should be discussed.
Reviewer 3 Report
The manuscript by Gonzales et al. presents an interesting adaptation of existing protocols for differentiation of astrocytes from NPCs. The only finding of this work is the observation that astrocytes differentiated from NPCs can adhere to non-coated plates without evident relevant impact on various aspects of their physiology. The authors suggest that this finding could yield cost benefits by reducing the costs of culturing NPC-derived cells (referred to as iAs, for iPSC-derived astrocytes). Following the establishment that iAs can attach and grow on non-coated vessels, the authors conducted an extensive set of experiments to assess the potential physiological effects on iAs, encompassing measures of proliferative rates, presence of astrocytic markers, glutamate uptake, activation by cytokine treatment, or mitochondrial function. This study holds promise for researchers involved in iPSC-derived astrocyte research, although additional efforts seem necessary to refine the results and provide further clarification on certain aspects of the findings. While acknowledging that this manuscript represents an important body of work, the reviewer notes that several details remain inadequately explained or described, imprecisions undermine the clarity of the text, and the omission of accounting for some differences acts as a barrier to its immediate publication. The primary objective of the revisions should be to effectively establish that astrocytes cultured over an extended duration on non-coated plates do not exhibit challenging physiological distinctions compared to cells cultured for an equivalent duration on coated plates.
Figure 1. The authors need to clarify how long the cells have exactly been grown before being cultured on non-coated plate. Stating that cells were “propagated in cAM for at least 60 days” is vague and not sufficient. The material and method section even stipulates that cells were expanded up to 110 days. The authors need to indicate the precise age of the cells (using notation such as DIV60, DIV110…). Cells cultured for different periods of time can reach different maturation levels, revealing or masking effects. At the very least, information about age discrepancies between each experiment must clearly be stated in the manuscript. One main question is: How long the cells have been in culture on non coated plates? This is crucial and unfortunately unclear. Only few days? Are 24h or 72h really enough for clear differences to be revealed? Answers to those first questions will clarify many aspects of the experiments presented.
The authors also need to clarify what is the exact composition of the media termed cAM. Astrocyte medium + supplements + FBS? Since it is complete, does it contain FBS and at which dilution? If FBS was used, was it always used? The scienCell medium could be used first with AGS and FBS to differentiate NPCs in astrocytes, then those astrocytes could undergo another step of maturation, in the same medium supplemented with AGS but no FBS. Was it the case? Astrocytes obtained in presence of FBS could be able to attach and grow, as they might retain some immature traits, but it remains unclear if more mature iAs obtained by a maturation step could also attach and grow on non-coated plates when FBS is absent.
One missing expected control is the growth of human primary astrocytes on coated and non-coated plates. As iAs are supposed to have properties very similar to human primary astrocytes, it could be interesting to test the effects of the absence of Matrigel coating on the cells that iAs are expected to replicate.
Figure 1A does not offer remarkably convincing images displaying the astrocytic features emphasized by the authors (same for S2A). What phase contrast method was used on the brightfield microscope ? The use of a cellMask staining could help the authors to better visualize the stellate shape and long cellular processes, and especially provide a way to quantify those features.
Figure 1C shows the exponential growth of cells with the cell number tripling within 3 weeks. However, figure 1E suggests that only 1-2% of cells express KI67 at the beginning of the experiment (day3 or 72h). After 3 weeks of recovery, are there differences between the coated and non-coated groups in terms of proportion of cells expressing KI67 (what the authors called percent expression)? Surely 1-2% of the cells being positive for KI67 at any given time cannot account for such a proliferation rate (tripling in 3 weeks). Hence, the proportion of KI67-positive cells after 10 or 15 days must be investigated and should be higher, possibly beginning to differ between the two groups. Same question for other cell lines. All graphs seem to indicate that cells need a period of time to restart growing at a constant rate, and 72h appears to fall on this acclimation period. Comparison with NPCs is not particularly relevant, statistical analysis between the two groups (non coated vs coated) are the relevant data that the authors need to perform systematically for all experiments, not only for figure 1. For instance perform comparisons of groups in S2D and S2G.
Figure2 also contains small mistakes that do not help the reading of the manuscript. Legend of figure 2B states a quantification of IF signal intensities, but the figure states that it is the percentage of cells expressing the given marker relative to DAPI. Those are two different things, which one is it? Moreover, the term “% positive cells relative to DAPI” can be replaced by the simpler “proportion of positive cells”.
The IF intensities appear very weak considering the important level of background. This doesn’t suggest a high expression levels for those markers, quite the opposite. The authors shouldn’t present just an overlap image (merge channels) of channels, this is difficult to evaluate. If it is really necessary to only show merge images, the use of colors and proper balance must be considered (is it red or magenta that was used? Because magenta is a color already containing blue). The weakness of intensities leads to wondering whether the signals are really specific.
Figure2B appears to indicate statistical differences between the two groups, regarding the number of cells expressing GFAP or S100B. Statistical differences must be calculated and commented on. Writing that they are similar is not enough. Culturing the cells for only 24h might counter-select for the healthiest cells, and the fixation should be performed after several days, when growth rates are constant (see comments above).
Figure 3 should present the RQ (2–∆Ct method) to appreciate the levels of expression of the markers relative to the endogenous marker. Are those levels high or very low? The text referred to figures 3G to K as experiments done on Axol cell line. But the legend refers to it as ATCC cell line. Which one is it? If it’s Axol, where are the ATCC data? Figures 3G to K data obtained in non-coated plate seems to be much more variable than coated plates. Does it reveal an inherent issue with those cells? An issue that could be much more evident after days or weeks of cultures rather than just 3 days?
In Figure 4A, the glutamate uptake could be normalized to total number of cells, or more easily, to the protein content. Such a normalization appears required to compare groups, even if the number of cells should be the same.
In Figure4B, it is unclear what is uncoated and coated, which is crucial. The observed age dependency for glutamate uptake should be discussed: could it be that the astrocytes are producing and releasing their own laminin or ECM and that over time, cells are adapting to the changing conditions around them. Could it be that when reaching a certain density, the cells are changing their physiology? That would happen in both groups, as figure4C suggests.
In Figure 4E, if the expression levels of the glutamate transporter EAAT1 is significantly higher without coatin: shouldn’t this have observable consequences on the glutamate uptake? Investigating the protein levels could address this question. Besides, isn’t a relevant information pointing out some differences between iAs grown in absence or presence of Matrigel? How long those cells were grown on non-coated plates? 3, 9, 14 days? Can the differences be abolished after several days, when ECM produced by astrocytes offer similar cellular environments in both groups? Time of culture must be clearer in the text or legends and protein levels investigated.
Figure4F careful examination suggests that iAs on non-coated plates might be more prone to react to Il1beta, but again statistical analysis is missing, even if the authors just write that differences are not statistically significant. Are error bars SD or SEM?
Figure4G. Is it the median density that is calculated or the average? This is not the same, and the choice for the median density over the average should be discussed. Are there statistical differences in 4J between groups? Another question that needs to be clarified and discussed based on calculated p values.
Figure5 The text says that there are no differences in non-mitochondrial respiration, while the legend says the opposite. Again, what are the p values calculated? With which raw data? The graph seems to show quite clearly a difference, another concerning example of a statistically significative difference interpreted by the authors as non-relevant. Same for the proton leak which is not “slightly higher” in coated plates as stated in the text, but appears to be more or less the double. And the authors minimize that observation. The authors should calculate parameters from ECAR experiments like they did for OCR. What is the glycolysis level? Glycolytic capacity? Glycolytic reserve? Same kind of questions can apply to data obtained with Axol where some differences are also observed.
Figure 6. Why EAAt1 difference in expression noticed in figure4E is not found in figure6?
Minor point
Line 143. Replace “Figs. 1D” by “Figs. 1E”
Round 2
Reviewer 3 Report
The authors provided sufficient corrections for the manuscript to be publishable.